# Study on FcγRn Electrochemical Receptor Sensor and Its Kinetics

**DOI:** 10.3390/molecules25143206

**Published:** 2020-07-14

**Authors:** Dandan Peng, Dingqiang Lu, Guangchang Pang

**Affiliations:** 1College of Biotechnology & Food Science, Tianjin University of Commerce, Tianjin 300314, China; Dazzle_Peng@163.com; 2Tianjin Key Laboratory of Food Biotechnology, Tianjin University of Commerce, Tianjin 300314, China

**Keywords:** γ-immunoglobulin (IgG), neonatal IgG Fc receptor (FcγRn), human FcγRn (hFcγRn) electrochemical receptor sensor, interconnected allosteric constants (Ka)

## Abstract

Neonatal γ-immunoglobulin (IgG) Fc receptor (FcγRn) is a receptor that transports IgG across the intestinal mucosa, placenta, and mammary gland, ensuring the balance of IgG and albumin in the body. These functions of FcγRn depend on the intracellular signal transduction and activation caused by the combination of its extracellular domain and IgG Fc domain. Nevertheless, there are still no kinetic studies on this interaction. Consequently, in the present study, we successfully constructed the human FcγRn (hFcγRn) electrochemical receptor sensor. The signal amplification system formed by chitosan nanogold-hFcγRn protein and horseradish peroxidase was used to simulate the cell signal amplification system in vivo, and the kinetic effects between seven IgG and hFcγRn receptors from different species were quantitatively measured. The results showed that the interaction of these seven IgGs with hFcγRn was similar to the catalytic kinetics of enzyme and substrate, and there was a ligand-receptor saturation effect. The order of the interconnect allosteric constants (K_a_), which is similar to the Michaelis constant (K_m_), was human IgG < bovine IgG < horse IgG < rabbit IgG < sheep IgG < donkey IgG < quail IgY. The results showed that hFcγRn had the strongest ability to transport human IgG, which was consistent with the evolution of the system. Therefore, our hFcγRn electrochemical receptor sensor can be used to measure and evaluate the interconnected allosteric network. It is also an essential parameter of the interaction between hFcγRn and different IgGs and, thus, provides a new detection and evaluation method for immunoemulsion, therapeutic monoclonal antibody therapy, heteroantibody treatment, and half-life research.

## 1. Introduction

The neonatal immunoglobulin (IgG) Fc receptor (FcγRn) is composed of a heavy chain α-chain with a molecular weight of about 50 kD and a light chain β_2_-microglobulin (β_2_m) with a molecular weight of about 15 kD. Its structure is similar to the heterodimer receptor of the major histocompatibility complex (MHC) class I molecule [1], while its main function is to mediate the non-specific endocytosis of IgG from pregnant and lactating mothers, which is transported to the fetus via the placenta barrier or to infants via the breast barrier and intestinal wall barrier so that pregnant and lactating mothers can transmit humoral immunity to offspring [2,3,4], as well as resist congenital diseases such as hypoproteinemia [5,6]. In addition, FcγRn is also a single receptor for IgG and albumin [7,8]. It binds to IgG/albumin in a pH-dependent manner (at acidic pH (pH ≤ 6.5) with a nanomolar affinity at a neutral pH or higher (pH ≥ 7.0) not combined or released) [9,10,11,12], so that IgG/albumin is internalized into the acidic endosome membrane where FcγRn is located by nonspecific endocytosis, which is free from degradation due to dissociation from lysosomal fate (Figure 1), thus ensuring the long half-life of IgG and albumin in vivo [13,14,15].

The two functions of FcγRn are mainly achieved through the interaction of multiple amino acid residues (such as tryptophan 131, glutamate 116, isoleucine 11, etc.) in α_1_ and α_2_ domains with CH_2_-CH_3_ amino acid residues (such as isoleucine 253, histidine 310 and histidine 435) in IgG Fc domains [3,13]. When IgG is released into the blood, IgG continues to bind to Fc receptors (FcRs), because FcRs of IgG are widely expressed in macrophages, monocytes, neutrophils, basophils, natural killer (NK) cells and other immune cells [16]. Therefore, when the antigen binds to the Fab end of IgG, FcRs generate positive or negative intracellular signals regulating cell response [17]. Some of the effector cells internalize the soluble immune complex or cell-antibody complex through endocytosis, and other effector cells kill the target cells through antibody-dependent cell-mediated cytotoxicity (ADCC) [18], which has great potential in the treatment of many diseases and even tumors by using therapeutic monoclonal antibodies. For instance, vaccinating pregnant women with influenza vaccine reduces the probability of infants to suffering from influenza and mothers to suffering from respiratory diseases [19,20]. Fc-insulin-encapsulated nanoparticles are used to target the intestinal mucosal transport of wild-type mice so as to render the wild-type mice in a long-term hypoglycemic state [21,22]. Generally, the function of the therapeutic monoclonal antibody is primarily determined by the interaction between the Fc domain of the antibody and the FcRs [23]. Therefore, quantitative evaluation of this interaction is essential for solving the half-life of therapeutic monoclonal antibodies however, only a few studies addressed this issue thus far.

Because the heterologous IgG can be transmitted to the blood circulation of the body through the intestinal mucosa and can be combined with various leukocytes with Fc receptor in the blood, it provides a theoretical basis for the passive immune function of milk. Substantial evidence has shown that the milk of one kind of animal can provide immune protection for its own children, but also for other different kinds of animals [24,25]. The heterogeneous transfer of passive immunity provides an opportunity for the development of immune milk. It was found that drinking specific immune milk can prevent rheumatoid arthritis, cholera, and cardiovascular diseases. For example, the widespread use of cow’s immune milk is considered to be a potential means of slowing down the outbreak of avian influenza, severe acute respiratory syndrome (SARS), and other human respiratory diseases [26]. At present, many products on the market, especially infant milk powder, claim to be rich in IgG. Nevertheless, it remains unknown whether IgG from different mammals can be transported to the human body through human FcγRn and not be discharged out of the body as a heteroantibody participating in immune protection.

When hFcγRn is combined with different concentrations of ligand IgG, different signals are input to the body, which generate corresponding output signals in the body. This causal relationship reflects the dynamic process of signal amplification generated by the interaction of hFcγRn and IgG after extracellular binding and transmembrane [27,28]. Therefore, in this study, the hFcγRn electrochemical receptor sensor was constructed, and the signal amplification system was formed by chitosan-gold nanoparticle-hFcγRn protein and horseradish peroxidase (HRP). The response current generated by the binding of IgG with hFcγRn measured by the electrochemical workstation at different concentrations, was used to simulate the process of signal transmission in vivo. Quantitative research was performed on the recognition, interaction, and linkage allosteric dynamics between different kinds of IgG and hFcγRn, with the aim of furthering the research on immune milk, therapeutic monoclonal antibodies, heterologous antibody treatment, and half-life.

## 2. Result

### 2.1. Ultraviolet–Visible Light (UV-Vis) Scanning and TEM Characterization of Gold Nanosol

Due to the high conductivity, large specific surface area, and negative charge on the surface of gold nanoparticles (GNPs), which can rapidly adsorb the IgG antibody with the Fc domain [29,30] under electrostatic interaction, GNPs were prepared to provide the electrical signal amplification effect on hFcγRn protein. The color of GNPs after preparation was wine red. The ultraviolet-visible light spectrophotometer was used next to scan the whole wavelength of the newly prepared GNPs in the visible light range of 450–700 nm. The results are shown in Figure 2. Transmission electron microscopy (TEM) was used to further observe the GNPs. The results are shown in Figure 3.

As shown in Figure 2, the configured GNPs reached the maximum absorption peak at 521 nm. The average particle size of GNPs corresponding to the absorption value was about 15–20 nm. As shown in Figure 3, the sphere diameter of GNPs was about 20 nm, and its shape was a regular sphere, with uniform particle size and no obvious agglomeration. Because the particle size of GNPs in the UV-Vis scan and TEM scan was basically the same, and GNPs were in favorable condition, the GNPs were successfully configured in this experiment. 

### 2.2. Electrochemical Characterization of the Assembly Process of hFcγRn Electrochemical Receptor Sensor

Using a 1 × 10^−3^ mol/L K_3_Fe (CN)_6_ solution (containing 0.20 mol/L KNO_3_) as the test bottom solution, the whole electrode assembly process was characterized by cyclic voltammetry with a scanning rate of 50 mV/s and a scanning range of −0.1 to 0.6 v. The characterization results are shown in Figure 4. Curves a and b show that the peak current value of curve b was much smaller compared to curve A. This was because the formed chitosan (chit) film blocked the electron transfer to the electrode surface, which proved that the chit film was successfully assembled. As shown in curve b and c, the redox peak current of curve c was much higher than that of curve b. This was because the GNPs of gold particles with a diameter of 1–100 nm had high electron density, specific surface area, and dielectric properties [29,30], which provided them with have excellent electron transfer ability and accelerated electron migration, thus indicating that the GNPs were successfully assembled. As shown in curve c and d, the redox peak current of curve d was smaller than that of curve c, because hFcγRn protein increased the resistance and blocked the electron transfer, which indicated that the hFcγRn protein was successfully assembled. As displayed in curves d and e, the redox peak current of curve e was larger than that of curve d. This was because the ability to promote electron transfer of thionine (thi) was stronger than that of chit, which indicated the successful assembly of the thi-chit complex. As demonstrated in curves e and f, the redox peak current of curve e was much smaller than that of curve f. This was because after HRP-GNPs were assembled, the redox ability of HRP significantly increased, which indicated that HRP-GNPs were successfully assembled. As shown in curve f and g, the redox peak current of curve f was much higher than that of curve g, which was because hFcγRn protein again prevented the electron transfer again, indicating that the second layer of hFcγRn protein was successfully assembled. As displayed in curves g and h, the redox peak current of curve g was lower than that of curve h. This was because bovine serum albumin (BSA) blocked the non-specific sites on the electrode surface, indicating that BSA incorporation was successful. In conclusion, the double gold-nanoparticles modified hFcγRn electrochemical receptor sensor was successfully assembled. 

### 2.3. Optimization of Determination Conditions of hFcγRn Electrochemical Receptor Sensor

#### 2.3.1. Determination of Constant Voltage

The sensor parameters of hFcγRn were measured using the time–current method. Since the binding of hFcγRn to its ligand IgG changes the signal, which could be evaluated by the change in response current determined by electrochemical workstation [27], and since the change of this combination process takes a specific time, we used time–current method to measure the sensing parameters. We used ultrapure water as the blank control (to eliminate the interference of impurities. The change rate of steady-state current before and after the combination of 1 × 10^−8^ mol/L human IgG solution and hFcγRn with respect to the voltage was used to measure the effect of different voltages on the electrochemical response of the sensor. As shown in Figure 5, when the voltage was −0.38 V, the change value of response current was the largest; thus, −0.38 V was selected as the constant voltage of the electrochemical receptor sensor of hFcγRn to measure different IgG solutions.

#### 2.3.2. Determination of Data Collection Time 

The stable current change rate of 1 × 10^−8^ mol/L human IgG solution under different binding time with respect to the different binding time was used to determine the time when the stable current value was stable after the combination of hFcγRn and IgG. Furthermore, the data collection time after ligand–receptor interaction was determined. As shown in Figure 6, the change rate of the signal current of the electrode after 20 s basically remained unchanged. Therefore, the time that could stabilize the current value for 20 s was the data collection time when the hFcγRn electrochemical receptor sensor interacted with different IgG solutions.

### 2.4. The effect of IgG on hFcγRn Receptor

#### 2.4.1. The Interaction between IgG and FcγRn Receptor 

The electrode was assembled using the method outlined in Section 4.3.3. The minimum detection value of each IgG was determined using the method outlined in Section 4.5, and the response current of seven IgGs was tested to determine their concentration range. The logarithm value of IgG concentration with respect to the change rate of response current was determined, as shown in Figure 7.

As demonstrated in Figure 7, the change rate of response current of different kinds of IgG in a certain concentration range showed a favorable linear relationship with the increase of in IgG concentration, as follows: bovine IgG ranged from 10^−22^ to 10^−20^ mol/L, goat IgG ranged from 10^−21^ to 10^−19^ mol/L, human IgG ranged from 10^−23^ to 10^−21^ mol/L, and rabbit IgG ranged from 10^−21^ to 10^−18^ mol/L, horse IgG ranged from 10^−22^ to 10^−20^ mol/L, donkey IgG ranged from 10^−21^ to 10^−19^ mol/L, and quail IgY ranged from 10^−19^ to 10^−17^ mol/L. Therefore, we could detect the current change rate of different kinds of IgG in a smaller concentration range corresponding to different IgGs. 

#### 2.4.2. Kinetic Interaction Curve of IgG and FcγRn Receptor 

The range of IgG detection concentration, which had a good linear relationship with FcγRn, was further subdivided. The concentration of IgG measured with respect to the change rate of current was determined. At the same time, we used origin 9.0 software to perform the hyperbolic fitting for each curve, as shown in Figure 8. 

As shown in Figure 8, at a lower concentration, the change rate of current increased with the increase of in ligand concentration, indicating that hFcγRn was not saturated at this time. When the concentration reached a certain level, the change rate of current was very low or even constant with the continuous increase of ligand concentration, which indicated that hFcγRn reached saturation. The R^2^ of the curve of the current change rate of the seven IgGs was measured in the corresponding concentration range beyond 95%, which showed that the action of the seven IgGs and hFcγRn conformed to the hyperbolic law. 

According to the receptor-ligand binding, the kinetics equation was as follows:(1)$$\left[R\right]+\left[L\right]\overset{\;k_{1}\;}{\underset{\;k_{2}\;}{⇌}}\left[\mathrm{RL}\right]\to \mathrm{Electrochemical}\;\mathrm{signal}\;\mathrm{change}$$
When the receptor was saturated, the Equation (2) was obtained:(2)$$Ka=\frac{k_{2}}{k_{1}}=\frac{\left[R\right]\left[L\right]}{\left[RL\right]}$$

When [RT] is the initial concentration of the acceptor, then [R] = [RT] − [RL]. When [LT] is the total ligand concentration, then [L] = [LT] − [RL]. [L] = [LT] − [RL] and [R] = [RT] − [RL] are substituted into Equation (2) to yield:[RL]^2^ − [RL]{[RT] + [LT] + Ka} + [RT][LT] = 0(3)

The above equation is a hyperbolic unary quadratic equation with [RL] as a variable. When [RT] and Ka are fixed, [RL] alters in accordance with the change of [LT]. It rapidly increased in the early stage, and gradually plateaued in the middle and late stages. It was shown that the interaction between the receptor and the ligand had a ligand saturation process, whereby Euation (3) shows the saturation equation of receptor-ligand interaction. This process was similar to the substrate saturation effect of the interaction between the enzyme and the substrate, and the substrate saturation effect was a sign of the catalytic kinetics of the enzyme.

Upon modification of Equation (3), Equation (4) was obtained:(4)$$\frac{1}{\left[RL\right]}=\frac{1}{\left[RT\right]}+\frac{K_{a}}{\left[RT\right]}\frac{1}{\left[L\right]}$$

The above equation is a double reciprocal equation with 1/[RL] as the dependent variable and 1/[L] as the independent variable. The figure of this double reciprocal equation is a straight line. The slope of the line is K_d_/[RT], the intercept at the *x*-axis is −1/K_d_, and the intercept at the *y*-axis is 1/[RT]. Therefore, this study could be used to evaluate the interaction kinetics of hFcγRn with ligand IgG by using the constant (K_a_) similar to the Michaelis constant (K_m_) in this equation. In this study, this constant (K_a_) is called the interconnected allosteric constant.

#### 2.4.3. Linkage Allosteric Constant of the Interaction between IgG and hFcγRn Receptor 

According to the above derivation, based on Figure 8, the Lineweaver-Burk method (also called the double reciprocal method) was used to draw a double reciprocal curve (Figure 9) of seven IgGs within a certain concentration range. The abscissa of the graph was the reciprocal of the subdivided seven IgG concentrations, and the ordinate was the reciprocal of the corresponding current change rate. It could be seen from Figure 9 that the value of the linear regression equation, the correlation coefficient (R^2^), and the interconnected allosteric constants (K_a_) generated by the interaction of IgG with hFcγRn in a certain concentration range are listed in Table 1. 

The interconnected allosteric constants (K_a_) and the Michaelis constant (K_m_) are similar. The value of K_a_ is similar to that of K_m_, which is defined as the ligand concentration when the receptor saturation effect is half reached because the K_m_ value of an enzymatic reaction is negatively related to its catalytic effect. Therefore, a smaller K_a_ value denotes stronger interconnected allosteric dynamics between the ligand and receptor. As shown in Table 1, the order of K_a_ after the interaction of seven IgGs and hFcγRn was human IgG < bovine IgG < horse IgG < rabbit IgG < sheep IgG < donkey IgG < quail IgY. Obviously, the K_a_ value produced by the interaction between human IgG and hFcγRn was the lowest, which indicated that human FcγRn has the strongest ability to operate human IgG. After the interaction of quail IgY and hFcγRn, the K_a_ value of quail was the largest, which indicated that human FcγRn has the weakest ability with respect to quail IgY operation. The result of the K_a_ value was basically consistent with that of system evolution, which shows that the result of this experiment was reliable. 

### 2.5. Determination of Specificity, Stability, and Reproducibility of hFcγRn Electrochemical Receptor Sensor

#### 2.5.1. Specificity of hFcγRn Electrochemical Receptor Sensor

The response current of 10^−23^ to 10^−20^ mol/L human IgG solution and 10^−19^ mol/L human IgG solution containing the same volume of three kinds of interferences (the concentrations of IgA, IgM, and IgE solution were all 10^−6^ mol/L) was measured using the hFcγRn electrochemical receptor sensor. According to the change rate of the response current, it could be judged whether the interference had an effect on the combination of IgG and hFcγRn. As shown in Figure 10, when these four kinds of interferences were added to the 10^−19^ mol/L human IgG solution, the change rate of response current after stabilization did not decrease significantly, which suggested that the hFcγRn electrochemical receptor sensor had high specificity. 

#### 2.5.2. Stability of hFcγRn Electrochemical Receptor Sensor

The hFcγRn receptor sensor was continuously measured in 1 × 10^−8^ mol/L human IgG solution 10 times. The relative standard deviation (RSD) of the current change rate was 6.48%, which showed that the stability of the sensor was favorable. The batch of hFcγRn receptor sensors was stored in phosphate-buffered saline (PBS) solution at 4 °C, and the same concentration of human IgG solution was measured every day. The results are shown in Figure 11, where the change rate of response current sharply decreased from the ninth day. This process showed that the hFcγRn electrochemical receptor sensor could be stored in PBS buffer at 4 °C for at least nine days.

#### 2.5.3. Reproducibility of hFcγRn Electrochemical Receptor Sensor

One of the five batches of receptor sensors prepared in this experiment was taken out, and the same batch of receptor sensors was taken as a group, while the same concentration of IgG solution was detected in each group. As shown in Table 2, the RSD of the response current change rate (ΔI/%) was 4.91%, which indicated that the reproducibility of the receptor sensor was good.

## 3. Discussion

Over recent years, many methods were proposed to determine the interaction between IgG or Fc fusion protein and FcγRn (as shown in Table 3). The binding affinity between the extracellular domain of FcγRn and the IgG Fc domain is usually used to show the interaction. Yet, the function of cells in the body to transport IgG across the placenta or intestinal mucosa through hFcγRn not only depends on the affinity after binding but also on the signal transduction and the cascade amplification dynamics of the signal after binding, i.e., the interconnected allosteric effect [27,28]. Therefore, it is necessary to develop a method to detect the dynamics of FcγRn and IgG in vivo. In this study, the hFcγRn electrochemical receptor sensor was successfully prepared. The current change caused by the combination of IgG and hFcγRn was measured by the signal amplification system formed by chitosan nanogold–hFcγRn protein and horseradish peroxidase, which achieved the determination of the kinetic parameters of the interaction between IgG and hFcγRn. Because the sensor simulates the cell membrane adsorption process of the hFcγRn receptor in vivo, and because the peroxidase signal amplification system simulates the cell signal amplification system in vivo, the measurement results of this method are closer to the results of the intracellular signal transmission process after the action of ligand–receptor in vivo. 

Coincidentally, Guangchang Pang et al. [32] also studied the interaction between different IgGs and hFcγRn, and the types of IgG they used were basically consistent with the ones used in the present study. Although their research results also proved that heteroantibodies could enter the body through hFcγRn located on the surface of intestinal mucosa to transport from the lumen to the blood flow, and then through the blood to the innate immune cells containing FcγRs to exert immune protection, their results were based on the binding affinity of the extracellular domain of FcγRn to the IgG Fc domain. This study was based on the analysis of the allosteric dynamics produced in the cells after binding. Moreover, the results of Guangchang Pang et al. [32] showed that hFcγRn had the strongest affinity for rabbit IgG, while the results of this study showed that hFcγRn had the most potent kinetic transport effect on human IgG. Obviously, there was a big difference between the two results, with our results being closer to the system evolution. This further shows the reliability of the hFcγRn electrochemical receptor sensor. It provides a quantitative detection method for prolonging the half-life of IgG dissociation from lysosomal degradation, related to the therapeutic monoclonal antibody action on the body. Furthermore, the transport function of hFcγRn to bovine IgG is second only to that of human IgG, followed by horse IgG, rabbit IgG, sheep IgG, and donkey IgG, which provides a new detection and evaluation method for the absorption of heteroantibodies added to immune milk by the human body and the treatment of heteroantibodies.

In this study, the kinetic parameters of hFcγRn to quail IgY were also determined. Numerous studies addressed IgY [37,38,39,40], because IgY is closer to the functional characteristics of IgG [37], and the research cost of IgY antibody is low, while the damage to animals is small [38,39]. However, there are few studies on the transfer of IgY through hFcγRn to the body and its immunoprotection. In this study, the parameters of the interconnected allosteric dynamics between quail IgY and human hFcγRn interaction were quantitatively evaluated. However, compared with the ability of hFcγRn to transport mammalian IgG in vivo, hFcγRn had the weakest ability to transport quail IgY, which provides a reference for the development of quail immune egg products, as well as a new detection and evaluation method for the development of more immune egg products and the research of IgY antibodies.

The hFcγRn electrochemical receptor sensor that was successfully prepared in this study had strong specificity, high sensitivity (up to 10^−11^ pM), good stability (stable storage for nine days), strong reproducibility (RSD of the response current change rate was 4.91%), and short detection time (the detection time from the detection of current value to the stability in the optimal binding time of 20 s was the detection time of this experiment, i.e., about 100 s), which further explains the feasibility of this study. Nevertheless, the present study had some shortcomings. Since the real combination of hFcγRn and IgG occurs in vivo, new methods are needed for improved evaluation and detection of the interaction between hFcγRn and different kinds of IgG.

## 4. Materials and Methods

### 4.1. Reagent

The goat IgG, horse IgG, human IgG, bovine IgG, and rabbit IgG were purchased from Beijing boas Biotechnology Co., Ltd. (Beijing, China) Donkey IgG was purchased from Southern Biotechnology Co., Ltd. (Nanchang, China), whereas quail IgY was purchased from alpha diagnostic Co., Ltd. (San Antonio, TX, USA) The molecular weight of IgG was 150 kDa. The recombinant heterodimer of human FCGRT/B_2_M was purchased from Sino biological company. (Beijing, China) Human FcγRn had a molecular weight of 43.5 kDa.

### 4.2. Equipment

The UV-1800 ultraviolet–visible light spectrophotometer was purchased from Shimadzu Instrument Equipment Co., Ltd. (Tokyo, Japan) The transmission electron microscope (Philips TECNAI G2F20 company), CHI660E electrochemical workstation, working glassy carbon electrode (Φ = 3 mm), reference Ag/AgCl electrode, counter platinum wire electrode was all purchased from Shanghai Chenhua Instrument Co., Ltd. (Shanghai, China) The ultrasonic cleaner was obtained from Kunshan ultrasonic equipment Co., Ltd. (Kunshan, China), while the electric blast drying oven was purchased from Shanghai Yiheng Science Equipment Co., Ltd. (Shanghai, China). An ultrapure water preparation machine was used Millipore Milli-Q pure water system. (Shanghai Yarong Biochemical Equipment and Apparatus Co., Ltd., Shanghai, China)

### 4.3. Preparation of hFcγRn Double-Layer Nanogold Electrochemical Receptor Sensor

#### 4.3.1. Preparation and Characterization of Gold Nanoparticle Sol

Firstly, the triangular flask, volumetric flask, and other glassware needed in the preparation process were cleaned and dried. Then, on the basis of the Frens preparation method [41], the gold nanoparticle sol was prepared by the reduction of chloroauric acid with sodium citrate. Then, 100 mL of 0.01 g/100 mL chloroauric acid solution was taken out and put into a triangular flask, while 4 mL of 1 g/100 mL sodium citrate solution was added and mixed well. The pH value of the preparation solution was adjusted to 7.0 with K_2_CO_3_ and Na_2_CO_3_ solutions, after which the lowest point of the concave liquid surface of the preparation solution was marked on the triangular flask. Next, the preparation solution was put in the microwave oven and heated for 15–18 min under low fire until the solution turned a bright wine red color, after which the heating was stopped. Upon removal, it was cooled at room temperature, ultrapure water was added to the scribe line, and the nanogold sol was obtained. It was stored in the dark at 4 °C. The UV-Vis spectrophotometer was used to scan the newly prepared nano gold in the visible light range of 450–700 nm. According to the relationship between the wavelength of the maximum light absorption peak and the diameter of gold nanoparticles in the nanogold sol, the size of the particles was preliminarily characterized by consulting the data. The shape, size, and dispersion of gold nanoparticles was further characterized by transmission electron microscopy (TEM).

#### 4.3.2. Pretreatment and Characterization of Glassy Carbon Electrode

A proper amount of aluminum powder (α-Al_2_O_3_) was placed on the suede, a small amount of ultrapure water was dropped, and the glassy carbon electrode on the suede was polished using an eight-shape approach. The polished electrode was placed in the ultrasonic water bath for 30 s, after which it was repeatedly washed at least three times with ultrapure water. The cleaned electrode was placed in 1 mol/L H_2_SO_4_ solution and scanned with cyclic voltammetry to activate the electrode. The scanning range was −1 to 1.0 V, while the scanning rate was 100 mV/s. The activated electrode was placed in 1 × 10^−3^ mol/L K_3_Fe(CN)_6_ (containing 0.2 mol/L KNO_3_) solution. The electrode pretreatment effect was characterized by cyclic voltammetry at a scan rate of 50 mV/s and a scan range of −0.1 to 0.6 V. The electrode could be used only when the peak potential difference of the cyclic voltammetry curve of the pretreated glassy carbon electrode was less than 80 mV. If not, the above steps were repeated to reprocess the electrode. The electrode was taken out after scanning. It was washed with ultrapure water and put in a nitrogen environment to dry for subsequent use.

#### 4.3.3. Assembly Method of hFcγRn Electrochemical Receptor Sensor

Six microliters of 0.5% chitosan solution was taken out (dissolved chit with 1% acetic acid solution) and was dropped onto the surface of the glassy carbon electrode. Then, it was dried in a drying oven at 37 °C for 30 min. After the chitosan of the electrode core formed a film, the electrode was taken out and was placed on a super clean workbench to cool to room temperature. The cooled electrode was immersed in 0.5 mol/L NaOH solution for 5 min, rinsed with ultrapure water, and put in ultrapure water for 0.5 h. After drying the above electrodes, they were placed in the pre-prepared nanogold sol and self-assembled at 4 °C for 24 h. The electrode was taken out and washed with ultrapure water several times, after which 10 μL of hFcγRn protein solution was added to the surface of the electrode, and the self-assembly of the electrode was continued at 4 °C for 24 h. After the electrode was taken out, it was washed with ultrapure water and dried. Next, 5 μL of Corydalis chitosan complex (2.5 mL of 2% chitosan solution (2 g of chitosan solution in 100 mL of 1% volume ratio acetic acid solution) + 320 μL of 10% glutaraldehyde solution + 200 μL of 0.01 M Corydalis solution + 2% acetic acid solution) was dropped in the center of the glassy carbon electrode and dried at room temperature. The surface of the electrode was cleaned with ultrapure water and dried, after which the electrode was placed in the solution of nanogold horseradish peroxidase (1 mL of nanogold mixed with 1 mL of 2.0 g/L HRP, which was left to stand in the refrigerator at 4 °C for 12 h) for self-assembly at 4 °C for 24 h. The surface of the electrode was washed with ultrapure water, and then the electrode was placed in hFcγRn protein solution at 4 °C for self-assembly for 24 h. The electrode was taken out and washed with ultrapure water, then soaked in BSA solution at 37 °C (0.5 g/100 mL) for 1 h in order to block the non-specific sites. After sealing, it was taken out, washed with Tris-buffered saline (TBS) solution, and naturally dried to obtain two layers of nanogold-modified hFcγRn electrochemical receptor sensors. The sensor was stored in PBS buffer at 4 °C. The relevant assembly process is shown in Figure 12.

### 4.4. Determination of Different Kinds of IgG Using hFcγRn Double-Layer Nanogold Electrochemical Receptor Sensor

In this experiment, the working electrode, the reference electrode, and the counter electrode three-electrode system was used for determination. The working electrode was the assembled hFcγRn electrochemical receptor electrode, the reference electrode was the Ag/AgCl electrode, and the counter electrode was a platinum wire electrode. Ultrapure water was used as a blank control. the response currents of different IgGs interacting with hFcγRn receptors were measured using the amperometric I-T method under the optimal voltage. 

The electrode reaction of the measurement process is based on a series of redox reactions produced by 8 mmol/L H_2_O_2_ in the test substrate and HRP and Thi (the determination of the H_2_O_2_ concentration refers to the previous work of the research group [42,43]). the specific reaction process is as follows:HRP + H_2_O_2_→HRP(ox) + H_2_O(5)
HRP(ox) + 2e^−^ + 2H^+^→HRP + H_2_O(6)
HRP(ox) + Thi(red)→HRP + Thi(ox)(7)
Thi(ox) + 2e^−^ + 2H^+^→Thi(red)(8)

In equations, HRP(ox) represents the oxidized horseradish peroxidase, and Thi(red) and Thi (ox) represent the reduced and oxidized thionine respectively. During detection, a double substrate reaction (H_2_O_2_ and thionine) actually occurs on the electrode, where reaction (7) and reaction (6) proceed simultaneously, thereby accelerating the conversion of HRP(ox) to HRP. The response current of the prepared electrode should theoretically be the sum of the current of reaction (8) and the current of reaction (6).

When the FcγRn receptor binds to its ligand-IgG, the FcγRn-IgG complex formed on the electrode surface could hinder the electron transfer due to the change of steric hindrance, which in turn causes a weak electrical signal change. This change in electrical signal can be transmitted to the electrochemical workstation through the signal amplification system, allowing us to determine the change in steric hindrance based on the change in current. Therefore, in this study, the change rate of the response current before and after the binding of IgG to FcγRn can be used as the detection index of IgG concentration. The calculation formula was as follows:(9)$$\Delta I/\%=\frac{I_{1}-I_{2}}{I_{1}}\times 100$$
where *I*_1_ and *I*_2_ represent the steady-state current value of ligand IgG at the same time point before and after measurement.

Ligands transmit environmental signals through cell membrane surface receptors. When the receptor of FcγRn binds to its ligand (IgG), even if there are multiple interferences, it can recognize micro IgG [27,28]. Yet, this binding is a molecular recognition process, which mainly relies on intermolecular forces such as hydrogen bonds, van der Waals forces, and ionic bonds. Because the recognized FcγRn can be saturated by the ligand IgG, and IgG with different concentrations has different molecular numbers, the interaction strength differs [27]. There are numerous articles that used the electrochemical biosensor of the receptor assembly signal amplification system to measure the response current generated after ligand binding to the receptor to simulate the interaction of receptor–ligand binding in vivo [44,45]. Therefore, in this experiment, the current range of different concentrations of IgG solution interacting with hFcγRn was used to determine the concentration range of IgG solution. This range was subdivided to obtain the kinetic curves of the interaction between different IgGs and hFcγRn. Finally, the interaction between hFcγRn and different types of IgG was evaluated through the kinetic curve of the interaction between different IgGs and hFcγRn, and the interconnected allosteric constant (K_a_) of hFcγRn was calculated using the double reciprocal method.

### 4.5. Data Analysis

The Original 9.0 software was used to analyze and process the current change data measured by the sensor. The lowest concentration of ligand IgG that could be detected was taken as the limit of detection (LOD), and the concentration of the electrical signal that responded to the current was three times higher than the standard deviation of the blank control signal (signal-to-noise ratio (S/N) = 3).

## 5. Conclusions

In this study, an FcγRn electrochemical receptor sensor was constructed. The signal amplification system formed by chitosan–gold nano–hFcγRn protein and horseradish peroxidase mimics the in vivo cell signal amplification system. The interaction of different kinds of IgG and hFcγRn was tested. Among the measurement results, human FcγRn showed the strongest ability to operate human IgG, which confirmed the successful preparation of this sensor. Furthermore, the detection method was simple, specific, sensitive, stable, reproducible, and with a short detection time. This provided a new detection and evaluation method for immunomilk, monoclonal antibody therapy, heteroantibody therapy, and extended half-life research. To the best of our knowledge, this is the first study on the development of the FcγRn electrochemical receptor sensor.

## Figures and Tables

**Figure 1 molecules-25-03206-f001:**
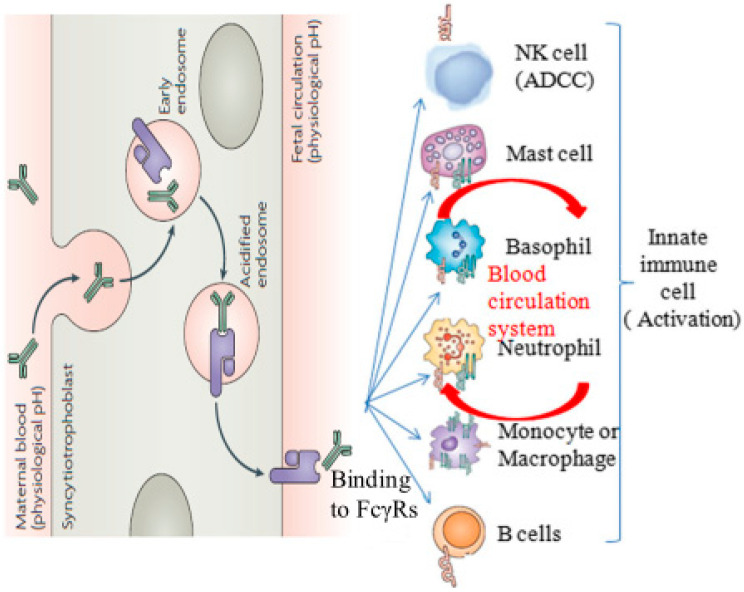
Fc receptor (FcγRn)-mediated immunoglobulin G (IgG) transfer in the human body. FcγRn in intracellular vesicles allows IgG to enter the cell through non-specific liquid endocytosis, and then internalizes in an acidic environment where the endosome of FcγRn is bound to FcγRn. After that, under the action of physiological pH, FcγRn transports IgG out of the cells and reaches the innate immune cells containing FcγRS to play out the immune role through the blood.

**Figure 2 molecules-25-03206-f002:**
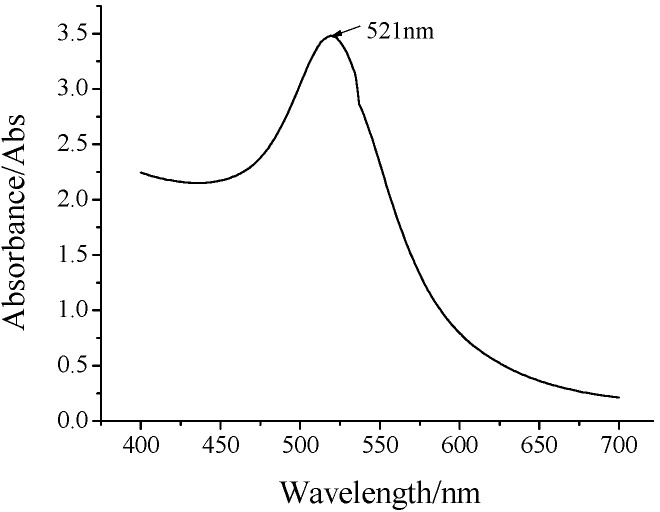
Ultraviolet–Visible Light (UV-Vis) scan images of gold nanoparticles (GNPs).

**Figure 3 molecules-25-03206-f003:**
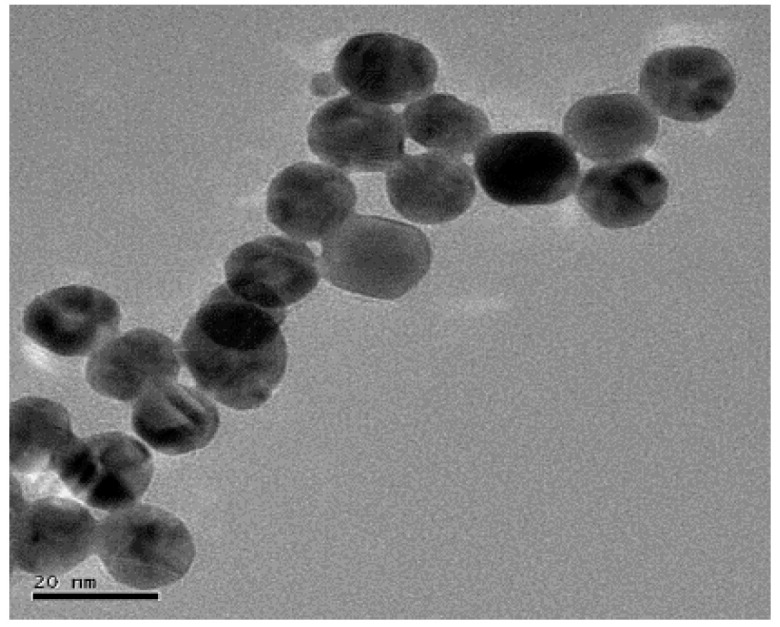
TEM representation images of GNPs (145,000×).

**Figure 4 molecules-25-03206-f004:**
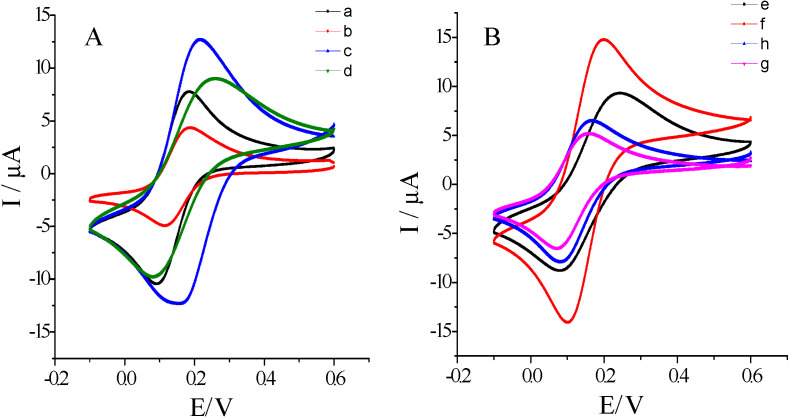
Assembly process of cyclic voltammetry curve of hFcγRn electrochemical receptor sensor. (**A**) (a~d): (a) Bare electrode (glassy carbon electrode (GCE)). (b) Chitosan (chit–GCE). (c) GNPs–chit–GCE. (d) hFcγRn protein–GNPs–chit–GCE. (**B**) (e~h): (e) Thionine (thi)–chit–hFcγRn protein–GNPs–chit–GCE. (f) Horseradish peroxidase (HRP) –GNPs–thi–chit–hFcγRn protein–GNPs–chit–GCE. (g) hFcγRn protein–HRP–GNPs–thi–chit–hFcγRn protein–GNPs–chit–GCE. (h) Bovine serum albumin (BSA)–hFcγRn protein–HRP–GNPs–thi–chit–hFcγRn protein–GNPs–chit–GCE.

**Figure 5 molecules-25-03206-f005:**
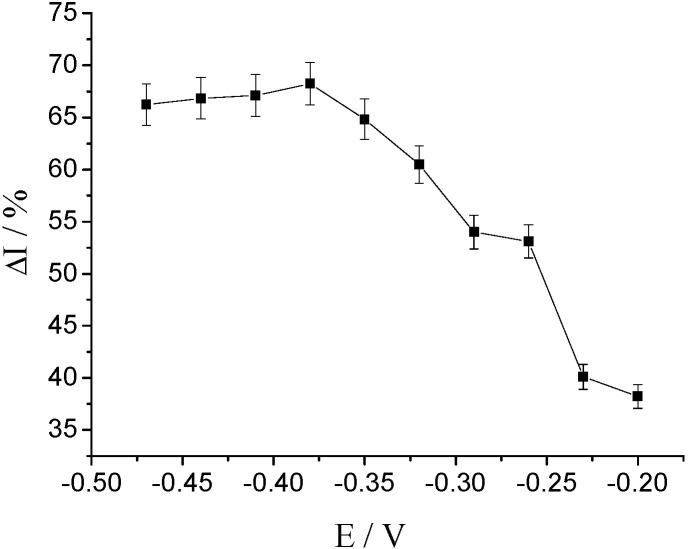
The effect of different potentials on the response of hFcγRn receptor biosensors. The detection potentials were −0.2 V, −0.23 V, −0.26 V, −0.29 V, −0.32 V, −0.35 V, −0.38 V, −0.41 V, −0.44 V, and −0.47 V.

**Figure 6 molecules-25-03206-f006:**
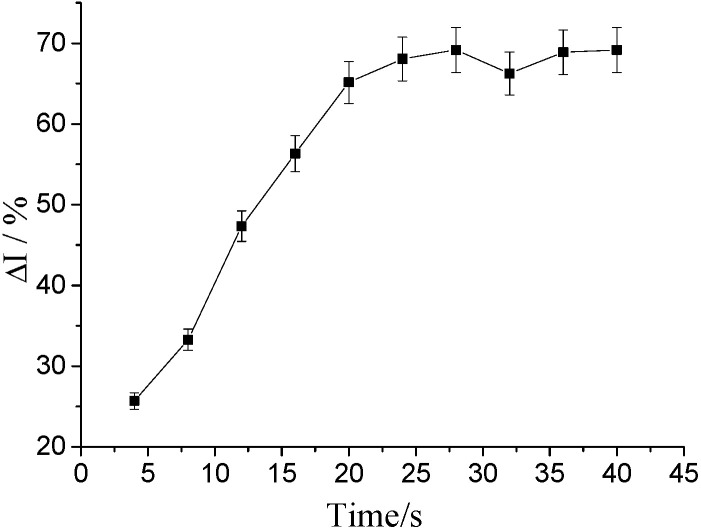
The effect of different binding times on the response of hFcγRn receptor biosensors. The detection times were 4 s, 8 s, 12 s, 16 s, 20 s, 24 s, 28 s, 32 s, 36 s, and 40 s.

**Figure 7 molecules-25-03206-f007:**
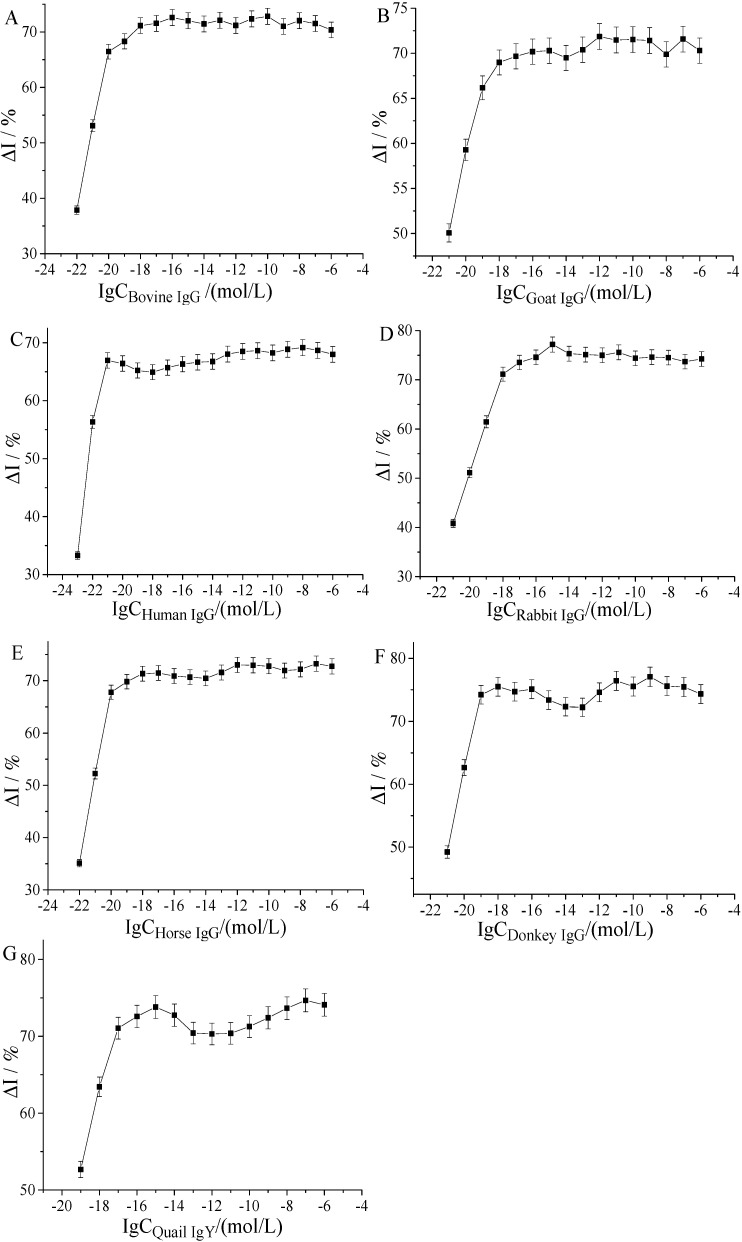
The current change rate of seven IgGs in the detection range. Bovine IgG (**A**), goat IgG (**B**), human IgG (**C**), rabbit IgG (**D**), horse IgG (**E**), donkey IgG (**F**), and quail IgY (**G**). All data show the mean ± SD of three measurements.

**Figure 8 molecules-25-03206-f008:**
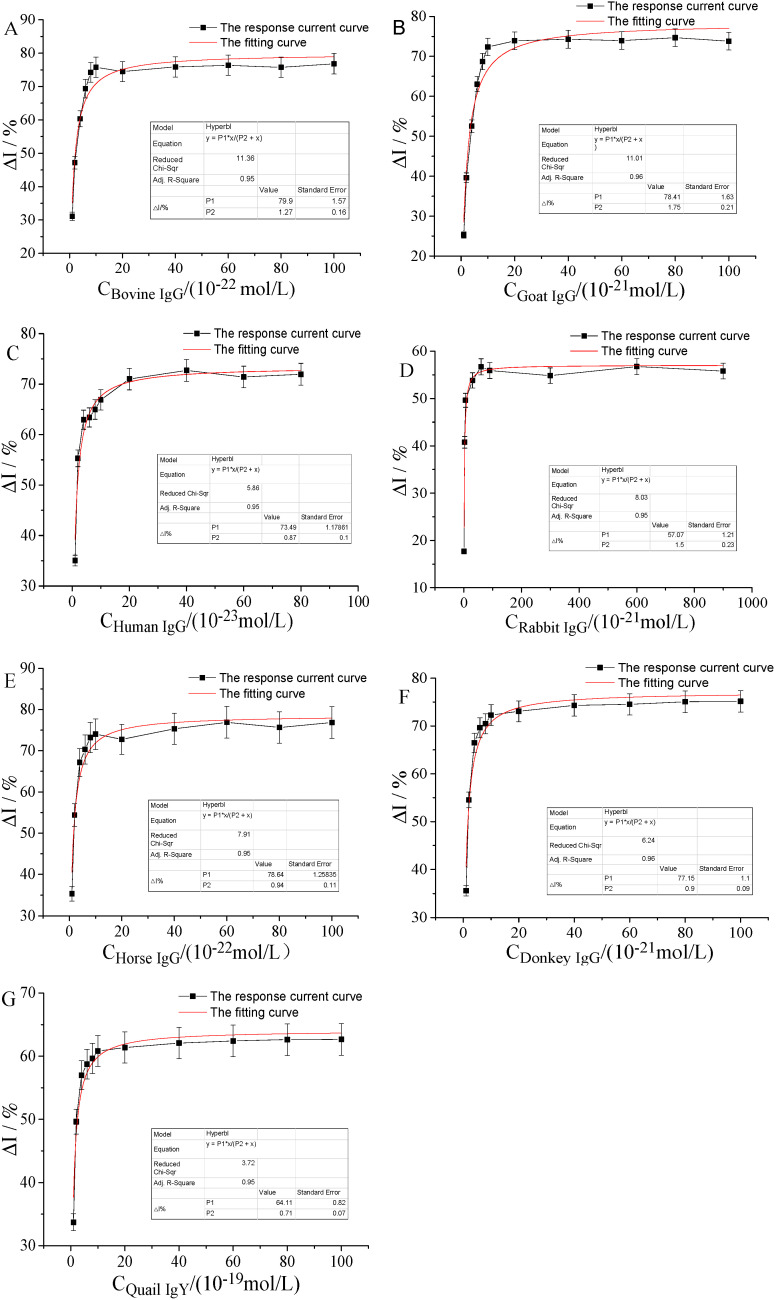
The current change curve and corresponding fitting curve of seven IgGs detected in a certain concentration range. (**A**) Bovine IgG 10^−22^ to 10^−20^ mol/L, (**B**) sheep IgG 10^−21^ to 10^−19^ mol/L, (**C**) human IgG 10^−23^ to 10^−21^ mol/L, (**D**) rabbit IgG 10^−21^ to 10^−18^ mol/L, (**E**) horse IgG 10^−22^ to 10^−20^ mol/L, (**F**) donkey IgG 10^−21^ to 10^−19^ mol/L, and (**G**) quail IgY 10^−19^ to 10^−17^ mol/L.

**Figure 9 molecules-25-03206-f009:**
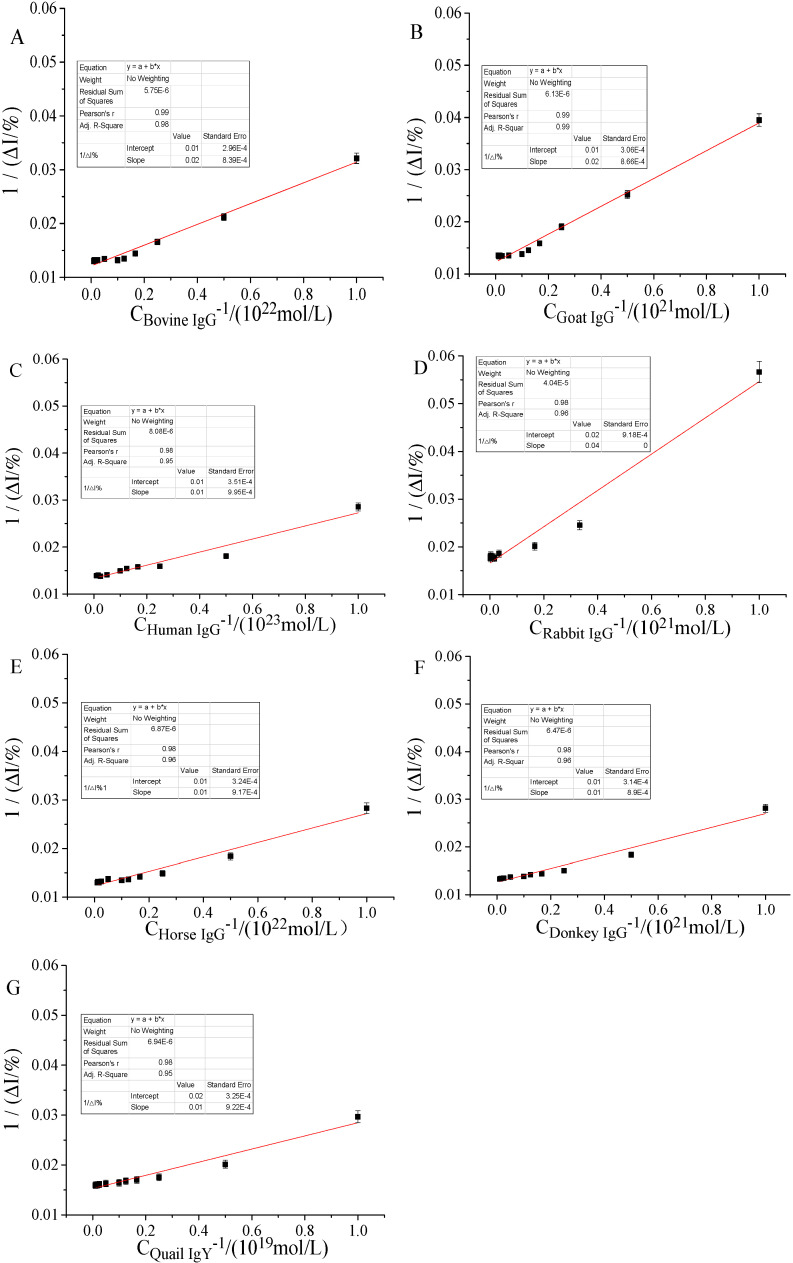
Double reciprocal curve of seven IgGs detected in a certain concentration range. (**A**) Bovine IgG 10^−22^ to 10^−20^ mol/L, (**B**) sheep IgG 10^−21^ to 10^−19^ mol/L, (**C**) human IgG 10^−23^ to 10^−21^ mol/L, (**D**) rabbit IgG 10^−21^ to 10^−18^ mol/L, (**E**) horse IgG 10^−22^ to 10^−20^ mol/L, (**F**) donkey IgG 10^−21^ to 10^−19^ mol/L, and (**G**) quail IgY 10^−19^ to 10^−17^ mol/L.

**Figure 10 molecules-25-03206-f010:**
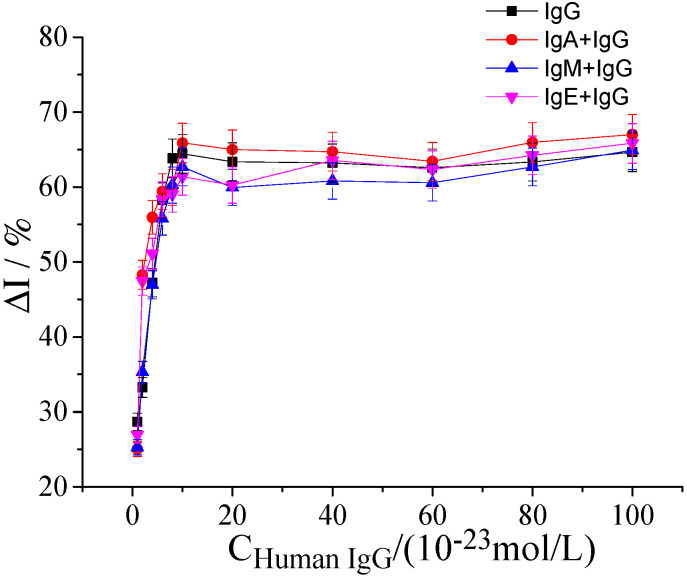
The change rate of response current of hFcγRn electrochemical receptor sensor to IgG, IgA + IgG, IgM + IgG, and IgE + IgG. The concentration range of IgA in IgG solution was 10^−23^ to 10^−20^ mol/L, and the concentrations of IgA, IgM, and IgE in the same volume were 10^−6^ mol/L.

**Figure 11 molecules-25-03206-f011:**
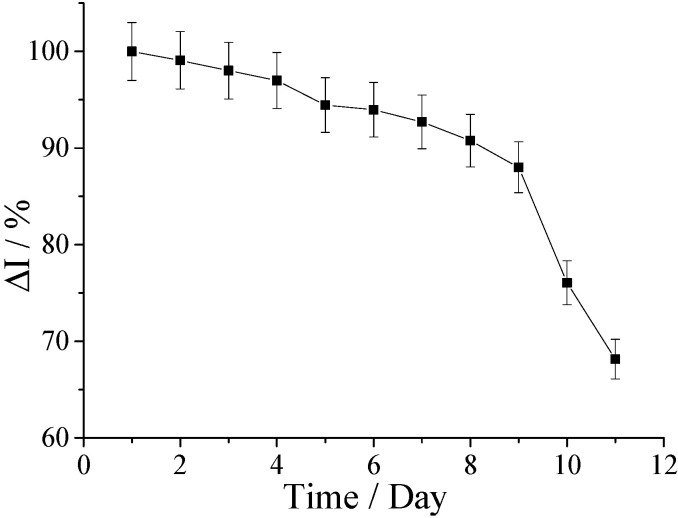
Inspection of sensor stability.

**Figure 12 molecules-25-03206-f012:**
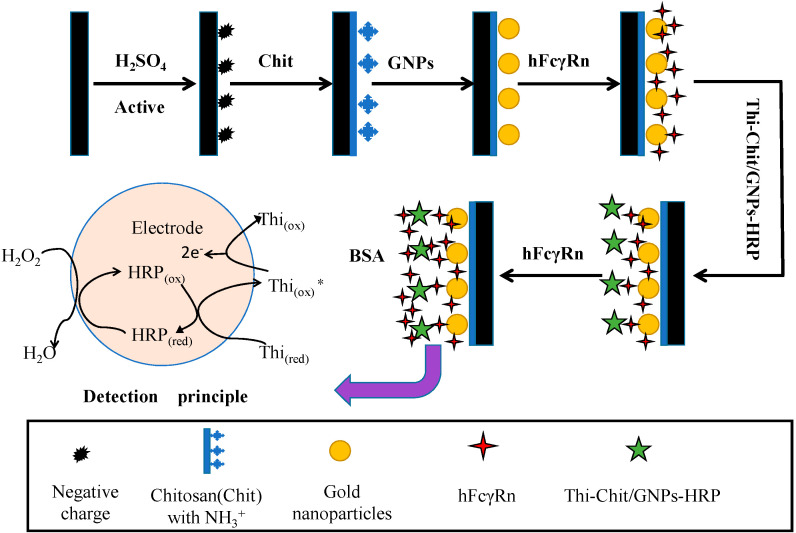
Preparation of hFcγRn receptor sensor.

**Table 1 molecules-25-03206-t001:** Linear regression equation, *R*^2^ and K_a_ of seven IgGs in a certain concentration range after interaction with hFcγRn.

Types of IgG	Linear Regression Equation	*R* ^2^	K_a_/(mol/L)
Bovine IgG	$\frac{1}{\Delta I/\%}=0.0193\times 10^{-22}\frac{1}{C}+0.0121$	0.98	1.59 × 10^−22^
Sheep IgG	$\frac{1}{\Delta I/\%}=0.0267\times 10^{-21}\frac{1}{C}+0.0123$	0.99	2.17 × 10^−21^
Human IgG	$\frac{1}{\Delta I/\%}=0.0139\times 10^{-23}\frac{1}{C}+0.0133$	0.95	1.02 × 10^−23^
Rabbit IgG	$\frac{1}{\Delta I/\%}=0.0139\times 10^{-23}\frac{1}{C}+0.0133$	0.96	2.29 × 10^−21^
Horse IgG	$\frac{1}{\Delta I/\%}=0.0148\times 10^{-22}\frac{1}{C}+0.0123$	0.96	1.20 × 10^−22^
Donkey IgG	$\frac{1}{\Delta I/\%}=0.0143\times 10^{-21}\frac{1}{C}+0.0126$	0.96	1.14 × 10^−21^
Quail IgY	$\frac{1}{\Delta I/\%}=0.0131\times 10^{-19}\frac{1}{C}+0.0153$	0.95	8.61 × 10^−20^

**Table 2 molecules-25-03206-t002:** Experimental results of sensor reproducibility.

Numbers of Sensor	Response Current (I_2_, μA)	Blank Current (I_1_, μA)	ΔI (%)
1	5.99	8.83	32.14
2	7.15	10.50	31.95
3	6.69	9.85	32.08
4	7.46	11.59	35.62
5	5.79	8.50	31.93

^1^ The response current (I_2_) and blank current (I_1_) of each group were the average ± SD of three detections in the group. ^2^ ΔI (%) was calculated using Equation (5).

**Table 3 molecules-25-03206-t003:** The detection method of interaction between IgG or Fc fusion protein and FcγRn.

Detection Methods	Time	Sensitivity	References
Asymmetrical flow field flow fractionation (AF4)	30 min	0.1 μM	[31]
Surface plasmon resonance (SPR)	30 min	10 pM	[32]
Flow cytometer	6 h	4.6 μg·mL^−1^	[33]
Biolayer interferometry (BLI)	11 min	4 nM	[34]
AlphaScreen	1 h	1.88 μM	[34,35]
Enzyme-linked immunosorbent assay (ELISA)	30 min	0.4 ng·mL^−1^	[36]
hFcyRn electrochemical receptor sensor	100 s	10^−11^ pM	This study

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
