# Peer review of "Study on FcγRn Electrochemical Receptor Sensor and Its Kinetics"

_molecules, 2020, doi:10.3390/molecules25143206_

Round 1

Reviewer 1 Report

Still some improvements are necessary. Please carefully reread, I still find typos.

203 205 Therefore, this study could be used to evaluate the interaction kinetics of hFcγRn with ligand IgG by using the interconnected allosteric constants (Ka) similar to the Michaelis constant (Km). 

This does not explain the difference. Please provide the mathematical expressions!

225 Typo similar. the value 

Table 2 please round your values. An accuracy up to 10-8 is suspicious, if I take your error bars into consideration. Please correct this.

Figure 9 Inserts, Please round the fit parameters in the inserts to reasonable values. 

Figure 9, please make all plots of the same scale. Run Y-axes from 0.01 to 0.06. Than the data will become comparable.

Author Response

Q1: Still some improvements are necessary. Please carefully reread, I still find typos.

203-205 Therefore, this study could be used to evaluate the interaction kinetics of hFcγRn with ligand IgG by using the interconnected allosteric constants (Ka) similar to the Michaelis constant (Km). This does not explain the difference. Please provide the mathematical expressions!

225 Typo similar. the value.

A: Thank you for your good suggestion. We have added mathematical expressions to explain the reason why this study could be used to evaluate the interaction kinetics of hFcγRn with ligand IgG by using the interconnected allosteric constants (Ka) similar to the Michaelis constant (Km). And the revised words were shown in red color in Section 2.4.2 of the manuscript. Please see page 11-12, line 193-214.

Q2: Table 2 please round your values. An accuracy up to 10-8 is suspicious, if I take your error bars into consideration. Please correct this.

A: Thanks reviewer. According to your suggestion, we have rounded all the values in Table 2 to uniformly retain 2 significant digits. All revised words were shown in red color (Page 16).

Q3: Figure 9 Inserts, please round the fit parameters in the inserts to reasonable values.

A: Thanks reviewer. We have rounded all the values in Figure 9 Inserts to uniformly retain 2 significant digits (Page 13).

Q4: Figure 9, please make all plots of the same scale. Run Y-axes from 0.01 to 0.06. Than the data will become comparable.

A: It is really true as Reviewer suggested. According to your suggestion, we have rivised all the Y-axes from 0.01 to 0.06 in Figure 9 (Page 13).

Reviewer 2 Report

The authors addressed all of the reviewer's suggestions and the article can now be accepted for publication.

Author Response

The authors addressed all of the reviewer's suggestions and the article can now be accepted for publication.

A: Thanks a lot for reviewer’s positive comments, suggestions and recognition.

This manuscript is a resubmission of an earlier submission. The following is a list of the peer review reports and author responses from that submission.

Round 1

Reviewer 1 Report

Please, rewrite your paper. What is your sample quality? What are your experimental setups and what kind of models do you take in order to interpret your data. 

In brief the authors try to uncover the affinity of different IgGs by their adsorption behaviour to gold nano particles. The method of choice was a UV-Vis spectrophotometer. 

Please give the concentrations in g/L and at least compare them to literature values.  

For what purpose do the authors introduce the Michaelis constant (Km)? To vaguely compare it to Ka. This does not explain Ka.

Line 89  includes Aurface Plasmon Resonance (SPR) change to Surface …
Line 103  The color of gold nanoparticles (GNPs) synthesized by Frens method [33] is wine red. The authors produced the gold nano particles themselves? How about the monodispersity of these nano particles.
Line 106/ 107 It can be seen from the data that the average particle size of the GNPs particles corresponding to the absorption value here is about 15-20 nm.... I do not understand this sentence.
Line 113 the GNPs is in good condition. The GNPs configuration was successful in this experiment. I do not understand these sentences

Line 117 These is a break in the the sentence.

Figure 7 and 8 should be put together, the models the authors fits are based on must be given, the minimisation algorithm must be mentioned.

How do the autors explain the different capacities in the adsorption data?

Please, do not change the scale of the x-axes, this runs the comparision of the different plots impossible.

Table 1 The fit parameters are low, the authors should switch from mol/l to g/l
Table 2 The digits for △I % do not match the error bars given in the figures, what is the difference between △I % and △I/%?

Quite frequent the sentences start or stop and leave this reader puzzled. 

Reviewer 2 Report

Intensive language editing is required (English grammar and typographical errors).

The authors should consider including the histogram graph in Figure 3.

It must be clear which voltammogram is showed in Figure 4. (first, second, …)

Caption of Figure 4 must be improved. It has much repetitions.

The results discussion must be improved. The authors only mention that they know or expect each result.

Please explain in the manuscript how the GNPs accelerates the electron migration. The addition of a proper reference is also required. “Because the excellent electron transfer ability of the GNPs accelerates the electron migration, indicating that the GNPs are successfully assembled.”

Why chronoamperometric experiments were performed? What is the time employed for data collection?

What is the meaning of x-axis in Figure 6?

How the limit of detection was calculated? Please describe it.

IxT experiments of Figure 6 must be presented in support information.

Figure 7 could be excluded since it is repeated in Figure 8. However, it is also not clear why the fitting of curves was performed without the error bars. Please clarify it and include the errors bars in Figure 8.

A table comparing the obtained results with ones from literature must be added.

Please explain why the authors have not presented experiments with interferents. Moreover, Elisa comparison would strengthen the method.

I suggest present the section 4 before the results section.

Correct ultra-pure to ultrapure.

The experimental section must be improved. The experimental parameters are poorly described, making it difficult for readers to replicate the results.

It is not clear the goals obtained in this study, since they mention that this research needs to be improved.

Reviewer 3 Report

In the submitted manuscript, Peng and coworkers reported a hFcγRn as the receptor protein to construct the hFcγRn electrochemical receptor sensor. 

The work is overall well organized and well written. The objective has been clearly stated and the conclusions are partially supported by the findings arising from the experimental part. Some points make the publication of this work premature at this moment:

  1. It can be better for the readers if the authors bring a shorter abstract.
  2. It can be better if the authors bring SEM images of the modified electrode in the manuscript.
  3. Page 5, line 158 – please bring the number of the measurements in the caption of the figure 5.
  4. Page 5, line 161 – please bring the electrochemical method that has been used in this part.
  5. Page 5, line 161 – please explain about determination of detection limit by details. What formula has been used for calculating detection limit? And how is the calculation?